# The Music of the Silent Exodus: Nunchi Bwa-ing and Christian Musicking in a Second-Generation Asian American Church

Kathryn Minyoung Cooke

Department of Music, Graduate School of Arts and Sciences, Columbia University, New York, NY 10027, USA; k.cooke@columbia.edu

**Abstract:** In 1996, Helen Lee dubbed the departure of second-generation Asian Americans from the non-English-speaking immigrant churches that they were raised in as the "silent exodus". This nationwide phenomenon was taking place largely because first-generation churches failed to provide the second generation with culturally relevant care that would enrich their ethnic, national, and spiritual identities. Glory, the church of focus in this study, was founded by and is home to many silent exiles. In hopes of being an enriching church for second-generation Asian Americans, pastoral staff and leaders have created spaces within Glory for racial identity and faith to be in conversation with one another. However, in regard to the music of the church, they were stumped on what could be done to make it uniquely and proudly Asian American. This conundrum inspired a key question in this study: What is distinct about the way that Asian Americans worship God through music? This study argues that the worship music at Glory Church is distinctly Asian American not by what is sonically perceived, but rather by what is physically performed and collectively experienced. The Korean-English, or Konglish, term *nunchi bwa-ing* (눈치 봐-ing) is utilized as a keyword to describes Christian musicking in a multilingual setting and foregrounds the Korean/Asian American worshiping body. This study concludes by looking forward and arguing that Asian Americans ought to amplify their worship music to the larger Contemporary Worship Music scene as it has the potential to be a powerful site of intergenerational healing.

**Keywords:** Asian American music; multilingualism; second-generation Korean American; Christian worship; Korean American church; Contemporary Worship Music; Asian American; embodiment; emotion

## 1. The Steven Yeun/Ali Wong Dichotomy

The dark comedy-drama BEEF created by Lee Sung Jin was one of Netflix's most popular shows in the spring of 2023. In addition to the acclaim that the series received, BEEF went viral for one particular scene in the third episode "I Am Inhabited by a Cry" S. J. Lee (2023) which featured Steven Yeun's character Danny having an emotional breakdown as he stood in a Korean American church congregation singing worship music. Yeun grew up attending a Korean American church and, although his experiences were not identical to those depicted in BEEF, being in a church setting singing with other Korean Americans helped him feel an intimate and personal connection with the emotional breakdown that his character underwent. Independent from the series, BEEF's church scene went viral because the accuracy of the scene, from the instrumentation and style of the music to the emotional intensity felt by everyone in the room, resonating deeply with people who grew up attending Korean or Asian American churches. Through its pinpoint accuracy, the scene became a catalyst for conversations concerning the cultural impact of the Asian American church in the lives of those who grew up in that religious environment.

While this scene carried an emotional weight for Yeun and other Asian Americans who grew up attending church, Yeun's co-star Ali Wong told him that she thought this scene was hilarious and found it even funnier that Yeun could not understand what was

so funny about it. What had been nostalgic and intimate recreations of Korean American church life for Yeun were caricatures that demonstrated how bizarre emotional expression is in worship settings for Wong. As much as many Asian Americans have experiences with and deep ties to the church and Christianity, just as many have never set foot in a Christian church and are thus unfamiliar with what goes on practically in a church service, let alone the role of worship music in those services.

When I began my research on worship music at Glory,[1] a church located in the heart of Manhattan, New York that is home to a predominantly young adult Asian American congregation, BEEF had not been released yet. After BEEF and the church scene rose in popularity, the series became an effective means to discuss a key question in this project: What is distinct about the way that Asian Americans worship God? When I interviewed Glory's founding pastor Rev. Dr. Choi, he brought up BEEF and drew a Steven Yeun/Ali Wong dichotomy as an illustration of Glory's Sunday service attendees:

> So now the question is, how do you reach out to a deconstructed Korean American Christian and at the same time, the Ali Wongs? One person's deeply moved and gripped by the music once again, the other person thinks this is a joke and hilarious. And that's sort of the tension that I find myself in because, as I mentioned before, my heart is for the Ali Wongs. Our heart is obviously for the Steven Yeuns that are de-churched, grew up in the Korean church left for five years, and are coming back or something like that. But it's also for the Ali Wongs. So how do we make it so that our church doesn't look funny or ridiculous? Intellectually credible, but at the same time, how do we emotionally also move like the Steven Yeuns?

Choi's goal to reach out and appeal to the Steven Yeuns and Ali Wongs does not come from a desire to merely stay relevant or be popular, but from Glory's mission statement: "inspiring thinkers to believe & inspiring believers to think".[2] The mission statement itself is not a uniquely Asian American one, but the predominantly Asian American demographic of the church since its founding in 2015 has challenged pastoral staff and volunteer leaders of the church to consider how to get Asian American non-Christian thinkers to believe and how to get Asian American Christian believers to think critically about their faith. Given Glory's ethnic composition and mission statement, pastoral staff and leaders have curated various aspects of the church to be distinctly Asian American to provide spaces within Glory for issues concerning race and Christianity to be in conversation with one another. Some of these initiatives have included organizing a church-wide retreat in September 2019 focused on race and faith, hosting a panel geared towards empowering Asian American women in light of the Atlanta spa shooting on 16 March 2021, and partnering with the Coalition of Asian American Pacific Islander Churches (CAAPIC).

However, when it comes to curating the music of Glory to be distinctly Asian American, my interlocutors among the pastoral staff and volunteer leaders admitted that, despite their theorizing and efforts, they were stumped on what could be done. According to my interlocutors, the music team (referred to as the praise team) leaders and pastoral staff reached two main conclusions after many conversations. The first was an acknowledgement that many of Glory's weekly service practices, from the music to the sermons, came from pervasively white schools of thought and that they needed wisdom in discerning what exists in Glory's Sunday service as a relic of white hegemony versus an expression of Christian faith.[3] The second conclusion was that any quick solution to add Asian sonic elements to the music such as to "kpop-ify" the service, add East Asian instruments, or sing in different languages, felt disingenuous to the identity of Glory.

The responses from my interlocutors made me wonder whether, despite the fact that Glory's praise team plays Contemporary Worship Music (CWM) songs written by mostly if not entirely by white artists, there was something beyond the sound of the music that made it distinctly Asian American. I drew inspiration from Wong (2004) who approached the ethnographies in her book *Speak it Louder* by analyzing the processes behind how Asian Americans make music rather than categorizing and specifying a comprehensive definition

of Asian American music. Within the emerging field of Christian Congregational Music, I looked to scholars who have written on body ethics and Christian musicking such as Steuernagel (2021) and Im (2021) by considering how worship music is practiced at Glory through the worshiping body as cultural expression. From my observations both on and off the church stage as praise team musician and regular congregant as well as through conversations with my interlocutors, I argue that the worship music at Glory Church is distinctly Asian American not by what is sonically perceived, but rather by what is culturally curated, physically performed, and collectively experienced. In this study, I use the Korean-English (colloquially known as Konglish) term *nunchi bwa-ing* (눈치 봐-ing) as a keyword to describe a uniquely Korean/Asian American act of Christian musicking. Given my argument that ethnic distinction in musical worship can have cultural potency through means other than sound, I conclude this paper by looking forward and arguing that Asian Americans can and ought to amplify their worship music by physically and emotionally engaging their culturally distinct worshiping bodies as it has the potential to be a powerful site of intergenerational healing.

## 2. Keywords and Positionality

Before I share the findings of my research, I wish to clarify my intention behind certain keywords and fields in this study. I refer to Glory as a second-generation Asian American church; however, the church itself is not formally labeled in that way, nor do all the attendees self-identify as second-generation Asian Americans. I choose to use this term to define Glory in order to acknowledge its historical positionality as well as the current composition of the church's population. Glory was founded by a group of 1.5/second-generation Korean Americans who were raised in a first-generation immigrant church, the story of which I will share in greater detail alongside the broader history of the Korean American church. Although the majority of Glory's Asian American population is Korean and the church's foundation is rooted in the story of the Korean American church, I refer to Glory as an Asian American church rather than Korean American in order to be inclusive towards the non-Korean Asian Americans who attend Glory. The pan-Asian nature of Glory's congregation is in itself an indication of its second-generation identity. Jeung (2005) argues in his book *Faithful Generations* that the tendency for Chinese, Japanese, Korean, and other Asian Americans to cross panethnic lines in the development of second-generation Asian American churches is evidence that racial dynamics play a crucial role in their lives as Americans. The panethnic unity in second-generation Asian American churches is built upon similar stories of oppression as a collective racial minority group as well as a communal belonging bound by similar networks and lifestyles. Therefore, referring to Glory as an Asian American church establishes it as a space of panethnic mingling where Asian Americans whose ancestors may have come from warring countries are able to attend church alongside each other.

The term second-generation Asian American, although meant to be used as a term of inclusivity and historical accuracy, still does not embody each person who attends Glory Church. Glory's current population consists of not only second-generation Asian Americans, but also Asians who come from different countries such as Canada, Singapore, Korea, Taiwan, and China. There are also non-Asian congregants, two of whom are white men on pastoral staff at the church. My interlocutors included members of Glory who came from each of these groups: Korean Americans, non-Korean Asian Americans, and White Americans. Given Glory's lack of self-identification as an Asian American church on both an institutional and congregational level despite the church's majority pan-Asian demographic, I asked each of my interlocutors the following question: Is Glory an Asian American church, and how is it/is it not? Each of my interlocutors needed to consider the question for a few moments before giving their own explanation about how the answer is not merely yes or no, but how the history, demographic, practices, teachings, and culture of the church complicate their response. As much as my interlocutors may be hesitant in labeling Glory as a second-generation Asian American church because of the confusion that they feel in

the intermingling of spiritual and racial identity, I believe that referring to Glory as such encompasses the very tensions that they face individually while navigating through their identity as a church collective.

As a second-generation Korean American, I felt a sense of belonging when I began attending Glory Church in late 2017. A few months later I joined the praise team as a multi-instrumentalist able to fill in a number of roles that the team needed such as performing on the cajon,[4] bass guitar, acoustic guitar, and singing. My experience on all these instruments was thanks to my own history learning the ropes on praise teams in Korean American church and youth ministry contexts. Given that I had experienced and practiced worship music in almost exclusively Korean American contexts, I only became aware of how deeply Korean American culture, particularly through multilingual practices such as Konglish, pervaded in Glory's praise team spaces when I began my research in 2022. One of my early observations, taken on a Sunday morning in November in which I was on cajon and the praise team and I were on standby waiting for service to begin, was pivotal in realizing how Konglish provides a framework to describe the worship music at Glory. I reflected the following in my field notes:

> Only a few people arrive early to service, two of which are a middle-aged (50s/60s) white couple who decide to sit in the third row in direct view of the praise team. There is now this sudden tension given how much this couple stands out in the room of mostly Asian Americans in their 20s–40s. This tension is quietly addressed in two ways. First, one of the church staff welcomes the couple personally, sits near them, and continues to make small talk with them. She wants to make them feel comfortable and they seem very touched by the gesture. Second, one of the praise team members asks under her breath, "What's this *baegin* (백인) couple doing here?" Baegin means white person in Korean and has a neutral connotation based on context. Here, the negative connotation is clear. No one on the team responds to her question, but her comment speaks volumes about belonging at Glory. Using the word baegin rather than "white" separates Koreans from non-Koreans. Everyone on today's team happens to be Korean, but there are many people in Glory's congregation who are not Korean.

Although Glory's Sunday service is held entirely in English, the conversations that take place among the congregants are multilingual in nature, creating a congregational environment that is deeply ethnically coded. Konglish is one of the most dominant languages spoken at Glory Church: as congregants speak to each other in English, they often find that there is an easier and more natural way to explain something using a Korean word. The use of Konglish is so prominent at Glory that there are often instances where non-Korean Asian congregants need to interrupt a Korean or Korean American to remind them that they are not Korean and could not understand the Konglish they just used. Some non-Korean congregants have encountered Konglish so often that they learn the meaning of certain Konglish terms and can understand the implication of the word, or even further, utilize it themselves.

As to the praise team member's use of Konglish, there are multiple ways to perceive her comment: as much as her use of Konglish could be interpreted as cultural exclusion, her actions could also be seen as a culturally sensitive means of dealing with the tension in the room. By using the word baegin, she hoped to specifically address the Korean praise team members and mitigate the confusion that she believed we also felt. Ideally, her use of Konglish would also be sensitive towards the white couple who was potentially in earshot of her statement. If they do not understand what she is saying, they will have no way of being discomforted by her question. However, there is no way of knowing for sure whether this white couple could understand her question or not. If the couple actually spoke perfect Korean or even just gathered context clues from the praise member's remark in order to understand her, her question would increase the tension in the room rather than reduce it.

In contrast, by quietly dissipating the tension in the room when the middle-aged white couple walked in, it is more appropriate to analyze the actions of the church staff person

who personally approached, welcomed, and started a conversation with them. This church staff person is also Korean American and her main responsibility is to direct the logistics of Glory's church services. Within the first minute of the couple walking into service and sitting down, the church staff person felt the same tension as the praise team member yet dealt with it in a way that included the couple rather than excluded them. As much as the couple stood out, the reality of Glory's congregation is that there are other middle-aged white members who call Glory their home, so it would not have been unheard of for this couple to feel comfortable at Glory. By personally welcoming the couple and engaging them in small talk, the church staff person made the couple feel at home which contributed to dispelling the tension in the room.

The church staff's observations, thoughts, and decisions can be consolidated into one action: she was exercising *nunchi* (눈치). Robertson (2019) defines nunchi as an amalgamation of intuition, perception, action, and social strategic maneuvering as well as "an expert, automatic skill in perceiving and deftly reacting to dynamic social and group situations, especially in ways that adroitly track (either intentional or unintentional) indirect communication" (Robertson 2019, p. 7). The misuse of nunchi, whether by under- or overutilizing it, is an example of cultural intimacy which Herzfeld defines as "aspects of an officially shared identity that are considered a source of external embarrassment but that nevertheless provide insiders with their assurance of common sociality" (Herzfeld 1997, p. 7). As Korean Americans, we were all aware of the praise team member's attempt at exercising nunchi, yet her misuse of it brought about an embarrassment and discomfort throughout the team. In terms of her misuse, the praise team member demonstrated an absence of nunchi by directly addressing the source of tension rather than maneuvering around it in a perceptive fashion. While there is a slight nuance of nunchi in her choice to use the word baegin rather than white, the discomfort caused by the question is apparent in the fact that no one on the team responded to her question despite all of us having understood it. Her question embarrassed the team because, by vocalizing her question in Konglish, she externalized our thoughts and exacerbated the very tension that she had hoped to absolve: our insider-ness versus the couple's outsider-ness.

Nunchi is a prevalent value in Korean American culture and the term has been adapted to fit into English sentences which has resulted in Konglish words such as *nunchi bwa-ing*. Adding the word *bwa* (봐) to nunchi turns the expression into an action that translates to "to look with nunchi". The present progressive -ing at the end turns the Korean term into an English verb, recontextualizing nunchi into Korean American culture. In stride with Fellezs (2019) in his book *Listen But Don't Ask Question* and how he uses olelo Hawai'i (Hawaiian language) keywords as a decolonizing exercise that centers and therefore mobilizes Native Hawaiian values, I use the Konglish term nunchi bwa-ing as a keyword to legitimize and empower Korean Americans and the way they communicate with each other in multilingual settings such as Glory Church. The utilization of Konglish words by Asian Americans[5] is an example of "ethnic invention" in which ethnic groups from immigrant backgrounds construct their own identity which strengthens the group itself thereby transforming and redefining what it means to be American (Conzen et al. 1992, pp. 4–5). Even when nunchi bwa-ing involves no language at all, the action itself is a byproduct that can only emerge from a multilingual community. When nunchi is properly exercised, it allows rituals to become spaces in which all participants can express, as Robertson puts it, "morally appropriate emotions" in "morally appropriate ways" (Robertson 2019, p. 26). In the case of musical worship at Glory as one of these said rituals, properly nunchi bwa-ing provides all participants, both Steven Yeuns and Ali Wongs, a morally appropriate space to worship God.

### 3. History of the Korean American Christian Church and Glory

During the Korean Civil War in the 1950s, American troops and missionaries came to aid South Korea which provided the conditions of possibility for a firm establishment of Christianity, particularly Protestantism. While Japanese and Chinese immigrants typically

came to the United States as single men in the early twentieth century, Koreans were able to immigrate as family units due to their affiliation with the Christian church and ministries (Hearn 2016; Yoon 2005). American missionaries and businesses assisted with their moves by combining vocation with religion. For example, many early Korean immigrants worked on plantations that were near a church that held worship services in Korean and also hosted English classes. Mark Chung Hearn writes that the Early Korean Church in America was "a site of solidarity and strength" (Hearn 2016, p. 67). When the Immigration Act of 1965 was passed which welcomed an influx of Asians of varying religious backgrounds to the United States, the Korean church was already established as an institution within Korean American society. There were therefore many Korean immigrant families who attended church for community rather than for religious beliefs because it was a safe haven where Korean culture and practices were preserved which Sang Hyun Lee, quoted by Helen Lee in 1996, refers to as "an inversion of status, a turning upside down of the way they are viewed in the society outside".

However, as much as these churches were safe havens for Korean American adults, the same could not be said of the children who were also attending the church. 1.5/Second-generation Korean American children[6] were not only being raised in the Korean church, but also in American schools and social settings which inculcated American ideologies and pressured them to learn English in order to properly assimilate. The Korean-speaking adults became aware of the discontent coming from the younger English-speaking congregants of the church and provided them with temporary solutions. Shin and Silzer (2016) refer to the most common temporary solution in their book *Tapestry of Grace* as the Room-for-Let.[7] The Room-for-Let is a small part of the immigrant church that is designated for English-speaking members, very often in the form of a youth or college group. Simply stated, it is a room or set of rooms within the immigrant church where first-generation Korean Americans let worship be held in English.

Within the Rooms-for-Let were Korean Americans holding services entirely in English which included the music that they sang as a congregation. Rather than singing from the Korean hymnbooks that the older generation used each week, the youth took inspiration from popular Contemporary Worship Music (CWM) groups and artists such as Hillsong Worship, Vineyard Worship, Chris Tomlin, and Shane and Shane for their worship music repertoire. The fact that acoustic guitar was the most common instrument a lead singer would use in CWM settings was conducive to the bare-bones structure of the Rooms-for-Let. Korean-style congregational worship, namely emotionally intense prayers that are shouted and cried out in congregational settings called *tongsongkido* (통성 기도)[8] still pervaded throughout the congregational musicking in the Rooms-for-Let, resulting in a new style of practicing CWM in a culturally distinct, second-generation Korean American matter.

As much as a unique second-generation Korean American way of worship was created in Rooms-for-Let, the Room-for-Let model was ultimately unsustainable because its only purpose was to keep the young congregants in the same church as their families. Shin and Silzer described the Room-for-Let model as a good immediate solution at best. Once these young English-speaking Korean Americans became adults, they would not be able to find appropriate resources for them within their Korean American church and would most likely look to another community to accommodate their needs. As this exact scenario unfolded for the first time in the 1990s, journalist H. Lee (1996) dubbed it the "silent exodus"—the departure of second-generation Korean Americans from the Korean-speaking churches that they grew up in because of the church's failure to attend to their needs.

The second-generation Asian American exiles followed different paths such as moving to a non-immigrant church, taking a break from church, or leaving the faith entirely. Regardless of their destinations, many of the silent exiles harbored hurt, bitterness, and jadedness towards their previous church experiences (Hearn 2016; Kim 2010; H. Lee 1996; Shin and Silzer 2016; Yoon 2005). Kim retells the story of a second-generation Korean American named Samuel who began to see the church as "an increasingly uncomfortable and oppressive place" as he was "trying desperately to disassociate from his Korean identity so that

he could 'fit in' with his mainstream white American friends" (Kim 2010, p. 1). Because of the "overly hierarchical and dictatorial" leadership paradigm that Samuel experienced at his church, he felt that he was unable to express his discontent which resulted in resentment and residual pain as he searched for a new church community (ibid.). Samuel, like many second-generation Asian American Christians, felt silenced and marginalized, which motivated his exile.

Glory's founding pastor Choi, who left his Korean immigrant church when he began college in 1997, pointed to a passionate movement within the silent exodus: "I would say, Asian Americans, as a whole in my generation, it was special. Everyone was going to missions, everyone was going to seminary. Everyone was becoming youth pastors, college pastors... that was almost the norm so much so that people almost felt pressured to". Shin and Silzer provide nine church models[9] that second-generation Asian American churches followed as a result of the silent exodus, two of which summarize Glory's history: the Church Plant and the Independent Church. The Church Plant often starts from scratch with a particular vision and mission to reach a specific group of people. In the case of Glory, Choi and four other founding members who were also Korean American began the church in 2014 with a vision to reach the Ali Wongs - skeptics of the Christian faith who did not grow up in the church. In terms of the music during the early days of Glory, Choi recalls the following:

> Our second service, because it was mostly skeptics, no one was singing. It was the least passionate church that I had ever been to... but I loved the fact that no one was raising their hands. I loved the fact that no one was singing and that no one was familiar with that stuff.

The Room-for-Let second-generation Korean American style of worship did not translate into the second-generation Asian American church that Pastor Choi had created. In fact, the majority of Glory's initial attendees were totally unfamiliar with what to do or how to behave during congregational singing, so they chose not to participate at all. However, the skeptics' lack of participation brought joy to Pastor Choi because he interpreted it as an indication that Glory Church was attracting the right type of people according to its mission. As of 2023, Pastor Choi continues to make it his goal to create space for non-believing skeptics at Glory, intentionally preaching to the congregation "as if they don't know anything". Being a safe church for non-believing thinkers to contemplate the Christian faith remains a primary goal for Choi and Glory Church.

Although Glory was created with Ali Wongs in mind, its culture and teachings also attracted Steven Yeuns—second-generation Asian American exiles who had grown up in the immigrant church and were searching for spiritual communities and resources that would bolster their faith and identity. One of my interlocutors theorized that silent exiles gravitated to Glory because "they [saw] the stories and messages that [were] coming from an Asian American pastor and [thought], 'he's saying that and I know why, I feel why, I understand why'". Within a few years, Glory's numbers quickly rose from around twenty attendees to over six hundred. Glory's growth cemented itself as an Independent Church which is distinguished from a Church Plant by its "size and fiscal strength" and "ownership of property" (Shin and Silzer 2016, p. 21). After using rented spaces from 2015 to 2022, the staff at Glory signed a 10-year lease in midtown Manhattan and moved into the space in January 2023. Glory's new worship space sits three hundred fifty people with a small stage where the praise team plays[10] and the pastor preaches. As Glory draws close to its tenth anniversary, Pastor Choi asserts that his vision for the church will stay consistent: "We still want to be a church for the lost people".

## 4. Liturgy and Musical Worship at Glory Church

Glory's musical worship is best understood within the totality of its weekly Sunday service. The order of the service, known as the liturgy, contains elements of verbal and nonverbal directions that prescribe an appropriate posture of worship/being for each portion of the service. Musical worship in the form of congregational singing led by a praise

band of volunteer members of the church consists of about a quarter of Glory's Sunday service. The band varies in its members on a weekly basis but will most often include a praise team leader who leads by singing and sometimes also playing acoustic guitar, one or two additional vocalists, a keyboardist, a bass guitarist, a cajon player, and a varying ensemble that might include a violinist, an electric guitarist, or both. As of 2023, Glory holds two services every Sunday at 10:30 a.m. and 12:30 p.m. and the service programs, including musical set and team, are identical to each other.[11]

Music is interwoven throughout Glory's liturgy as a means of both introducing the nonmusical elements of service that follow it as well as digesting what has preceded it. The instructions and directions given by the worship leader, pastor, and praise team leader predetermine and reinforce the type of posture that is deemed appropriate in order to maintain the ability for Glory's Sunday service ritual space to be one where, ideally, all people who attend can worship comfortably. The production and maintenance of ritual norms in Glory's Sunday services is saturated with nunchi bwa-ing. In the following section which delves into an example of musical nunchi bwa-ing at Glory, I focus on how the congregation nunchi bwa-s each other to standardize a physical posture of musical worship.

## 5. Nunchi Bwa-Ing at Glory

While nunchi bwa-ing plays an important role in maintaining the propriety of Glory's weekly ritual of Sunday service, nunchi bwa-ing needs to be done with discretion in musical worship. One of my interlocutors warns: "If you're always nunchi bwa-ing, you're always too afraid of offending your brother or sister—which I think is a dominant Asian American value, not being offensive". Robertson (2019) describes the overuse of nunchi as sycophantic and overly strategic, and I expand on this idea and interpret the overuse of nunchi in church music settings as compromising the quality of worship for those who desire to encounter God through music.

Given that nunchi bwa-ing translates to looking with nunchi, I went into Sunday service on a February morning and took on the role as ethnomusicologist prepared to look with nunchi, particularly curious about the bodily postures of the congregants as musical worship took place. I made myself comfortable in the fourth row although I usually enjoy being in the first row as it gives me more room to raise my hands during musical worship. By being in the fourth row, I knew that I would get a more realistic idea of what the typical Glory congregant would experience at Sunday service. As the service started and the music began to play, I glanced around the room at other peoples' postures and quickly came to the realization that I must not be the only ethnomusicologist in the room; in fact, there were observant ethnomusicologists all around me! Given my personal worship style, I usually am not attentive to how the people around me are worshiping; however, upon putting my ethnomusicologist hat on, I noticed that everyone was looking at each other. I even made eye contact with congregants which was quickly and awkwardly broken off by the other person. It occurred to me that all of these people were nunchi bwa-ing, glancing around as they gauged and deduced the ideal and most appropriate posture for Glory's musical worship. While a few people had their hands raised which has become a customary physical posture in Contemporary Worship settings (Lim and Ruth 2017), most people had concluded through their nunchi bwa-ing that a more appropriate posture for Glory's service was to have their arms lowered and, if they were to physically express any emotion, it should be limited to singing and relatively undramatic facial expressions. Many congregants, especially those who were newer to Glory or to any church setting in general, were nunchi bwa-ing each other so that the ritual appropriateness of Glory's Sunday service would be maintained; however, this decided-upon posture left me feeling unable to express my worship to God, and I wondered whether anyone else felt similarly. My experience with what I had interpreted as overutilization of nunchi bwa-ing at this particular Glory service led me to consider what unspoken negotiations were occurring in Glory's worship space that would result in such a reserved physical posture.

One of the most recent foci in the field of Congregational Music Studies has been placing congregational Christian music within the framework of ethics. In the fifth edited volume from the Congregational Music Studies Series titled *Ethics and Christian Musicking*, Myrick and Porter (2021) compile studies that highlight the ethical negotiations of identity, performance, and experience involved in curating musical worship for modern-day congregations around the world. The first section of the book, "The body and beyond", is of particular relevance to nunchi bwa-ing at Glory Church as Steuernagel (2021) retells the histories of congregational music, religion, and the body, as well as their fraught relationships with one another. Steuernagel argues that musical sound deeply engages the worshiping body and that Western Christian, more specifically White Anglo-Saxon Protestant (WASP), musicking body ethics have permeated congregational Christian music practices which places constraints on worshippers who wish to express their praise to God in more emotionally expressive ways. Im (2021) applies Steuernagel's arguments in her case study of Black gospel music performed by the Heritage Mass Chorus in South Korea. Given Korea's history of American missionization and Japanese colonization in the mid-twentieth century which resulted in modern day South Korea and its predominant Protestant religious atmosphere, singing Black gospel music provided the musicians of the Heritage Mass Chorus a space of catharsis and remembrance of their traumatic history in light of the Christian faith that they had embraced through it. Musically and physically speaking, the African American gospel tradition was a tool of reconceptualizing musical worship for the Heritage Mass Chorus by encouraging the use of chest voice and kinetic gestures such as kneeling, raising of hands, and clapping which was in direct contrast to the traditional Korean Protestant posture of worship and exaltation of a clean and pure voice (Im 2021, pp. 59–60). These simple changes in embodied musical worship expanded the range of Christian expressive possibilities for the members and leaders of the chorus.

I insert my study within the ongoing conversation of bodily ethics and Christian musicking, and particularly in the studies of Steuernagel and Im, by suggesting that the utilization of nunchi bwa-ing is an ethical negotiation that Asian American worshippers manage each week as they step into Glory's Sunday service. Each congregant's chosen posture of worship based on their nunchi bwa-ing is a physical manifestation of internal ethical negotiations of racial and national identity, religious performance, and congregational experience. The non-expressive physical worship stance that results from nunchi bwa-ing, while being comfortable for some congregants, compromises and constricts the worship experience for second-generation Asian American Christians who grew up in musical worship environments where physical and emotional expression were more encouraged and practiced.[12] Should nunchi bwa-ing be exercised as a means of creating a ritual space where Asian American congregants can experience a breadth of emotions in ways that are culturally and morally appropriate, Glory's musical worship could potentially become a critical site of emotional expression and healing for its participants.

Christian musicking as practiced in the West[13] has attempted to "erase the worshiping body" (Steuernagel 2021, p. 21). In studying congregational Christian music such as hymns, there has been a consistent overemphasis on analyzing the lyrics rather than the bodies who sing the lyrics. Additionally, the pursuit of piety was often equated to bodily self-control, or as Steuernagel puts it, "stillness over ecstasy" (Steuernagel 2021, p. 26). The worshiping body is a physical manifestation of internal ethical negotiations which wrestle with the norms set by Western Christian musicking traditions in which "Western Christianity's ethics straightjackets the body, which serves as the outward indicator of inward, spiritual realities. One might enjoy the music, but should not wear that enjoyment on the body" (Steuernagel 2021, p. 27). As the pastors and praise team leaders of Glory contemplate what aspects of their Sunday service are remnants of white hegemony versus expressions of Christian faith, they must also consider the worshiping body. In fact, they must not only consider the worshiping body, but consider the Asian American worshiping body. What aspects of stillness over ecstasy have seeped into the Asian American worshiping bodies at Glory, and how has that practice been exacerbated by the overutilization of nunchi bwa-

ing? The second-generation Asian American worshiping body is faced with a conundrum of having to suppress its inward ethnic identity and ways of expression for the sake of religious piety as constructed by Western Christianity. This conundrum creates challenges for Glory Church as they seek to self-determine how they will make music in their services given the worshiping bodies that are present at their church.

The second-generation Asian American worshiping bodies of Glory Church can be understood by considering the histories of Korean and Korean American worshiping bodies. In Steuernagel's chapter on bodily ethics, he references Nicholas Harkness's portrait of a Korean Christian singing a hymn in a theater in which she has "clasped her hands together, closed her eyes, and bowed her head in prayer" (Harkness 2014, p. 2). Steuernagel describes this posture as an example of the worshiping body under the influence of Western Christianity that pervades beyond North America and Europe. In the pursuit of physical piety in both musical tone and posture, Harkness's portrait of the Korean Christian is a "portrait of submission, an aspiration of immobility" (Steuernagel 2021, p. 31). Given the fact that American missionaries "installed a transpacific epistemic system in which Christians would continually conflate Euro–American cultural production with holiness", Korean Christians have come to associate Euro-classical vocal timbre and immobile musical posture with spiritual purity (Im 2021, p. 55). According to Im, the false association that Korean Christians have made between religious piety and the purity of vocal timbre and stoic posture has compromised the Korean worshiper's ability to process historical trauma in musical worship settings. She refers to the Western-influenced worshiping body in Korean Protestant musical worship settings as a posture that "exacerbates historical amnesia", erasing the national trauma that South Korea experienced as a Japanese colony as well as the U.S. American military presence throughout the Cold War period, which has resulted in its current postcolonial identity.

In stride with Im and her reflections on the timbre and body posture of Korean Protestant Christians, I suggest that the worshiping bodies at Glory strive for a similar pure sound and that its predominant worship posture resembles Steuernagel's conception of immobility. One of my interlocutors who is one of the praise team leaders commented on the style of music played at Glory Church which resonates with Im and Steuernagel's descriptions of straightjacketed Korean Protestant worship music: "The way we play... we value being really put together and clean and to a T versus more of a free style. That comes with the culture". He also commented on preferring certain types of voice styles to be lead singers on the team as one of the "very, very underlying things" that influence Glory's worship musical style:

> There are very, very underlying things, like the kind of singers that we consider to be good or that we like on the stage. The kinds of voices that we prefer are affected by the kind of music we listen to. As for Asian Americans, if you're really into kpop, you'll be into this kind of voice. It affects the team and its preferences.

The type of voice that my interlocutor is referring to when he mentions kpop is not so much mainstream kpop, but more of Korean singer-songwriter and acoustic pop[14] where singers' vocal timbres are recognized by their smooth, bright, and clean tones which are also characteristic of the timbres of Glory's praise team vocalists. Based on my conversations with vocalists on Glory's praise team, none of them claim to consciously manipulate the timbre of their voices while singing worship music, but recognize that their singing styles come from their musical backgrounds. One of the praise team leaders attempted to describe the musical backgrounds of a few vocalists on the team which ranged from "classically trained" to "acapella" to "kpop, ballad, and NRB".[15] These musical backgrounds embodied by the Asian American praise team members who vocalize them influence the praise team makeup and musical aesthetic of Glory Church. Upon other conversations with interlocutors and congregants, I often hear them describe Glory's worship music as clean, simple, and to the point. These descriptors are indicative of the fact that Glory's music, despite being performed thousands of miles away from Korea, resembles Korean Protestant worship musicking because of the second-generation Asian American worshiping bodies

who participate in it have been acculturated into a specific normative understanding of what it means to worship "correctly".

Due to the intergenerational pain that took place during the silent exodus in the 1990s, second-generation Asian American Christians have experienced a type of historical trauma of their own which, I argue, is being erased by allowing the mainstream WASP worshiping body to be the dominant posture of the congregants of Glory Church. Second-generation Asian American Christians grew up in their immigrant churches which became a "stressful bicultural context of balancing the oft-conflicting Asian parental and American cultural influences" (H. Lee 1996) which was a petri dish for internalized racism to germinate within the minds of silent exiles (Hearn 2016). This internalized racism has restricted second-generation Asian American Christians from seeing their worship to God as something unique and valuable, keeping them in a self-induced oppression in which White Anglo-Saxon Protestantism is the ideal paradigm of holiness and spiritual legitimacy. Similar to the Korean Protestants in Im's study, the second-generation Asian American Christians of Glory Church are experiencing their own historical amnesia which produces an attitude toward musical worship that aspires to immobility rather than a means of opening up a religious space of emotional expression and healing.

In his book *The Wounded Heart of God*, Park (1993) incorporates the Korean concept of *han* (한), a term for pain that encapsulates the suffering, trauma, bitterness, and guilt associated with being oppressed that Koreans and diasporic Koreans are intimately familiar with and feel united by, to Christian theology as a de-Westernized method of approaching sin and repentance. By reframing sin and repentance through the lens of han, Park urges that reconciliation must not only be between the sinner and God, but also needs to act upon the victims who have been sinned against. Second-generation Asian American Christians have been victims to a number of wounding events in their lives, and the historical trauma that their parents and grandparents experienced continues to live in subconscious memory. Second-generation Korean American Christians, for instance, carry not only the han of the silent exodus which was an amalgamation of feeling like a foreigner both in and outside of the Korean immigrant church, but they also carry the han of American and Japanese colonization that modernized South Korea but also motivated their parents to leave their homes and move to a foreign country in order to provide better lives for their families. Reconciliation from historical trauma must be a communal process that brings restoration to the silent exodus or, as Park writes, "Salvation is a relational event. It is a process of healing and freedom which transpires between sinners and their victims, and sinners and God" (Park 1993, p. 103).

## 6. There Is a Need for Healing

In April 2023, a guest pastor came to speak at Glory Church and showed a video from an NBC News (2016) series called *The Bridge* to demonstrate what it meant to seek God's face. The video, titled "Face to Face", featured young Asian American adults with their parents and each pair was instructed to look at each other without talking for four minutes. Each child/parent pair ended up crying because, in the four minutes where all they could do was gaze into the other person's face, both of them realized how much they loved each other and how often they avoided expressing that. Many of the parents commented through tears how, when they looked at their child's face, they became emotional because all they could see was the history of when their child was born to that present moment. Children, also through tears, talked about getting emotional at the thought of all the sacrifices that their parent made in order to give them the life that they currently have. The video clearly struck a chord with the congregation because, by the end of it, Glory's service space was filled with congregants sniffling with tears in their eyes. As the sermon wrapped up and the service continued in its typical liturgy of offering, closing song, and benediction, the sound of sniffles continued to reverberate throughout the space. Perhaps the sniffles were not actually reverberating, but I could not help but hear them so loudly and profoundly because this was the first time I saw and heard the people of Glory cry as a congregational body.

It was not a coincidence that what made the people of Glory cry was a video about the reconciliation of Asian American parents and their children. The tears and sniffles left a strong conviction in my heart after service: there is a need for healing.

Emotional expression in worship music through tears, raised arms, and more movement in the worshiping body can understandably come off as strange to someone who has never experienced that type of musical worship setting, let alone any type of musical worship setting or church service. Especially when a large part of Glory's mission statement is to appeal to non-believing thinkers, or Ali Wongs, who have never set foot in a church before and tend to carry a suspicion towards the Christian church and its affiliated institutions, it makes sense that certain musical choices of the church work towards not making its congregants uncomfortable or distracted during musical worship. However, when catering to church attendees who are not familiar with bodily expression in musical worship becomes an aspiration of immobility that is exacerbated by the overutilization of nunchi bwa-ing, second-generation Asian American Christians and their worshiping bodies become deprived of the opportunity to experience emotional healing through musical worship. If the leaders and pastoral staff of Glory church wish to critically consider what exists in their liturgy as remnants of Western/white hegemony, they must consider the worshiping bodies that participate in their services and the healing that they need.

Since its founding in 2015, Glory has become a church for the Steven Yeuns and Ali Wongs of Asian America to find culturally relevant resources, communities, and spaces that enrich their faith and identity. Nunchi bwa-ing is present at Glory Church as a tool of ethical negotiation in which congregants observe each other in order to determine the most appropriate physical posture of worship. The attitude induced by nunchi bwa-ing provides a uniquely Asian American way of worship that speaks to both the cultural specificity of the majority of Glory's congregants but also to the ways in which it articulates a distinctly Korean attitude toward social gatherings and, in particular, congregational worship. As my example of the church leader approaching the middle-aged white attendees indicates, nunchi bwa-ing allows for sensitivity in serving the Ali Wongs of Glory's congregation, who may not be familiar with an Asian American worship service. But nunchi bwa-ing may also be used to assuage the trauma experienced by the Steven Yeuns of the congregation by granting them the space to express themselves more ardently than adhering to WASP norms of musical worship. The people of Glory Church and the songs they sing are giving the silent exodus a voice that has the potential to heal current and future generations of the Asian American church.

**Funding:** This research received no external funding.

**Institutional Review Board Statement:** The study was conducted according to the guidelines of the Declaration of Helsinki, and approved by the Institutional Review Board of Columbia University (Protocol Code AAAU4904, approved on 1 February 2023).

**Informed Consent Statement:** Informed consent was obtained from all subjects involved in the study.

**Acknowledgments:** I would like to thank the members of Glory Church for offering their time, thoughts, and theorizing to this study. I would also like to thank the Music Department at Columbia University, especially Kevin A. Fellezs who advised this project in the form of a Masters Thesis. Finally, special gratitude goes to my friends and family who support me in all of my endeavors.

**Conflicts of Interest:** The author declares no conflict of interest.

## Notes

[1] By the request of the director of the church, I use pseudonyms for the church and my interlocutors in order to preserve their anonymity.

[2] Excerpted from the church's website. The mission statement is also printed on two banner stands on the left and right of the church stage and are displayed each week. Because of the constant presence of these banners, most church congregants are aware of the church's mission statement. It is also worth noting that many people are attracted to Glory Church because of its mission statement.

3   The scope of this paper will solely focus on the musical practices of Glory Church. For sources that elaborate on the broader negotiations within a second-generation Asian American church between Asian and white Christian practices, see Sharon Kim, *A Faith Of Our Own: Second-Generation Spirituality in Korean American Churches*.

4   Cajon has been the percussion instrument of choice at Glory Church ever since the praise team began incorporating more instruments into their ensemble aside from an acoustic guitar and vocalist. Cajon is generally a popular instrument in minimal and acoustic CWM settings.

5   Given the pan-Asian intermingling that happens in second-generation Asian American Christian settings, non-Korean Asian Americans have picked up on certain frequently used Konglish terminology and use them regularly, although not as often as Korean Americans.

6   1.5-generation means that the immigrant moved as a child and their move was more likely the choice of a parent or some other authority figure. The grouping together of 1.5- and second-generation Korean Americans in this sentence also goes to show how blurred these lines of migration are. I will continue to only use second-generation in this paper as a collective term for 1.5- and second-generation Asian Americans. It is worth noting that, for the sake of providing the necessary background history of Glory Church, I created two distinct groups of Korean-speaking first-generation immigrants and English-speaking second-generation immigrants. The reality of the history is that stories of immigration are always diverse in nature and there are many exceptions and blurring of lines to the narrative that I present in this study. For example, many first-generation Korean Americans can speak English but prefer to speak Korean, and the same goes for second-generation Korean Americans being able to speak Korean but preferring English. There are also young adult 1.5-generation Korean Americans who prefer to speak Korean, as well as older first-generation Korean Americans who are more comfortable communicating in English.

7   Shin and Silzer make it clear that the Room-for-Let model was drawn from Dr. Hoover Wong's book *Community: Coming Together or Coming Apart* which was self-published in 1999. In Wong's version the Room-for-Let was referred to as the Room-to-Let.

8   For more information on this type of prayer, see Yoon and his studies on *tongsongkido* (통성 기도) in "Christian identity, ethnic identity: Music making and prayer practices among 1.5- and second-generation Korean-American Christians".

9   The nine models from *Tapestry of Grace* are: Room-for-Let, Duplex, Triplex, Townhouse, Hotel, Church Plant, Independent Church, Satellite, and 2-in-1.

10  Praise team players on a stage facing the congregation is typical of contemporary Christian church spaces. For more information on space and contemporary worship, see Lim and Ruth Lovin' on Jesus (Lim and Ruth 2017).

11  The two services are identical with the exception of a separate children's service being available during the 10:30am service for parents to drop off their children at. The children's service includes a Sunday school where they are taught stories from the Bible.

12  In addition to the Room-for-Let musical culture that I describe in Section 2, second-generation Asian American youth ministries not affiliated with churches known as "para-church" ministries host musical worship events where emotional outpour and activities such as *tongsongkido* are encouraged and practiced. Some congregants of Glory Church, including myself, have been involved and even continue to be involved in these para-church events. Given the scope of this paper, I choose not to bring up para-church musical worship events despite their immense influence on second-generation Asian American Christians.

13  Steuernagel, upon writing that, "One might enjoy music, but should not wear that enjoyment on the body" notes in his paper that there are, of course, significant exceptions to this statement in Christian musicking as practiced in the West such as Pentecostal and charismatic expressions of Christian faith. He refers to these practices as, "exceptions that confirm the norm" Steuernagel (2021).

14  An example of a female kpop artist whose voice matches this aesthetic would be IU (아이유), especially in songs such as "eight" (2020) or "Ending Scene" (2017). A suitable example for a male artist would be Sam Kim (샘김), especially in songs such as "Love Me Like That" (2021) or "Sunny Days, Summer Nights" (2018).

15  NRB is a Konglish acronym that stands for *noraebang* (노래방), the Korean word for karaoke.

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
