# Peer review of "The Music of the Silent Exodus: Nunchi Bwa-ing and Christian Musicking in a Second-Generation Asian American Church"

_religions, doi:10.3390/rel15020244_

Round 1

Reviewer 1 Report

Comments and Suggestions for Authors

This is an excellent article. I have a few minor suggestions for making things clearer in the writing, included as comments in the attached file. It is almost ready for publication. 

Author Response

Thank you very much for taking the time to review this manuscript. Please find the detailed responses below and the corresponding revisions/

General - Block quotes have been edited, all minor edits were accepted and changed accordingly.

I added a few new paragraphs after reading other reviews such as detailing the music in the Room-for-Let (278-288), use of Konglish within the congregation at Glory, (175-185) and vocal timbre (506-522)

  1. Pg 3 - added the bold to clarify: “Given my argument that ethnic distinction in musical worship can have cultural potency through means other than sound, I conclude this paper by looking forward and arguing that Asian Americans can and ought to amplify their worship music by physically and emotionally engaging their culturally distinct worshiping bodies as it has the potential to be a powerful site of intergenerational healing.”
  2. Pg 5 - added this sentence to clarify: “As Korean Americans, we were all aware of the praise team member's attempt at exercising nunchi, yet her misuse of it brought about an embarrassment and discomfort throughout the team.”
  3. Pg 9 - I agree with specifying WASP earlier in the text, changes have been made accordingly.
  4. Pg 10 – I’m currently reluctant to make my introduction longer, but I’m taking your suggestion into consideration.
  5. Your comment: “This might not be useful for this paper, but in my experience these para-church organizations are where much more bodily involvement takes place for people who also attend mainline WASP churches as well, and it occurs to me that their existence might be allowing such churches to maintain this hegemony of immobility since their congregants are getting to worship in ways that are more expressive elsewhere?” I think about this all the time and would love to write about it more in the future!
  6. Pg 11 – I personally think that the next paragraph hits harder without a transition sentence. Same thought to before the conclusion – perhaps it’s a stylistic choice? I understand where you’re coming from though.
  7. Pg 12 – Thank you for pointing this out, that phrase was from a longer draft of this paper that talked more about theology!

Reviewer 2 Report

Comments and Suggestions for Authors

This is a great article - focused and well organized with clear descriptions. I marked "can be improved" because there are places where more detail would make it even more clear and evocative. Also, because you list "Contemporary Worship Music" as a keyword, you should include much more description of the music - see notes below.

Here are specific comments (numbers indicate lines unless indicated otherwise):

Note for all interview quotations - even if anonymous, they should be labeled (could be by pseudonym) and include date of interview

Effective opening - it draws the reader in and sets up the issues.

55ff: mark the quotation (indent)

103-4: “but rather by what is culturally curated, physically performed, and collectively experienced.” - come back to these phrases more explicitly throughout the article

108-110: “arguing that Asian Americans can and ought to amplify their worship music to the larger CWM scene as it has the potential to be a powerful site …” - I didn’t see this later in the article - expand/make it more explicit.

P. 3: suggest retitling heading #1 to include more information, such as “nunchi bwa-ing” instead of “keywords” and a phrase that describes the complex positionality rather than simply “positionality.” Better yet, a phrase that encapsulates how nunchi bwa-ing is related to or exemplifies positionality.

121 and surrounding: great explanation and justification of terms

165: set off/indent field notes

Page 4-5 - suggest bringing music into this story about white visitors - was any music being played, and if so how did it contribute to the feel of the room? If not, how did the silence potentially add to or reduce the tension? This will support the reference to musical practices in 238-40 (as it stands, that feels forced there).

First paragraph of p. 6 - add music to that history. What was music like in the churches of the 1950s and 60s? Second paragraph - were the “room-for-let” groups singing the same songs in different languages, or completely different repertoires?

301ff: this history could be mentioned in the discussion of Glory’s identity to bring add cohesion and context

306: indent quote - and add, what was that music that no one knew?

P. 8 under heading #3 - add more information: Are the songs projected or is there a book? If projected, describe the aesthetic of the slides (what type of font and colors; are there images?). How is the band amplified? In the earlier footnote about cajon you say it is often used for acoustic sounds; is the band amplified in a way that retains the acoustic sound, or are effects such as reverberation (and is the answer different depending on whether the electric guitar is included)? What are the timbres and effects used by the keyboard? Is the repertoire similar to Anglo and other churches (CCLI top 100, for example), any songs from Korea or other specific locations, or hymns from Korean or US or other contexts? 

355: “properness” sounds informal to me, suggest propriety or appropriateness, or another more focused word

365ff: use another word for ethnomusicologist at least one of these times so it’s not overused - participant observer, person engaged in fieldwork? (this will also help clarify to readers unfamiliar with what an ethnomusicologist does). Also in this section, explain more about what nunchi bwa-ing is doing/producing in this context - as a musician on stage, you had probably seen that many people aren’t raising their hands; was this the first time you noticed people glancing at each other? Is that the main evidence of nunchi bwa-ing here, or could you point to other evidence? Also, distinguish this process from the individually internalized ideas about posture/use of the body. As I understand it, this is the heart of your argument (that nunchi bwa-ing is the in-the-moment negotiation that brings together all the historical complexity in the context of a community), so draw out the explanation. Add more of your own observations/descriptions of people at Glory throughout your discussions of Steuernagel and Im in this section in order to bring their observations into fine detail and into your words. You should include (at least) as much analysis/description here as you did for the first explanation of nunchi bwa-ing (the reaction to the white couple), including how the same process can lead people to different actions - in this case, for some to raise their arms and others to have other physical postures. You could also include more about the music here - do some songs/types of sounds more clearly lead to different postures? How does the music interact with the nunchi bwa-ing?

415-428 plus fn 10: it would be helpful to have an orientation to this tension in the use of the body in Asian American worship earlier. In one of the intro/history sections, add a couple sentences about the different historical influences that affect how the body is used in worship today. Fn 10 could go earlier along with that.

Conclusion - Add an example of how music has allowed for healing: 542-544 expand the description of the music - you focus on the sniffling, but the space in the music allowed for those emotions to be processed and felt. You could also explain how this happened either despite or because of nunchi bwa-ing and how this might inform future uses of music for healing.

Author Response

Thank you very much for taking the time to review this manuscript. Please find the detailed responses below and the corresponding revisions.

General - Block quotes have been edited, all minor edits were accepted and changed accordingly.

I added a few new paragraphs after reading other reviews such as detailing the use of Konglish within the congregation at Glory (175-185)

  1. Labeling quotations – Can you give me an example of how to do this as an in-text label? Is there a way of doing it without using a pseudonym, and just by referring to the date of the interviews?
  2. Include more description about the music – I added two new paragraphs that describe the music more – one is in section 2 (278-288) and the other is about vocal timbre (506-522).
  3. 108-110 - Adjusted to better fit the conclusion: “…I conclude this paper by looking forward and arguing that Asian Americans can and ought to amplify their worship music by physically and emotionally engaging their culturally distinct worshiping bodies as it has the potential to be a powerful site of intergenerational healing.”
  4. Retitle Heading #1 – I agree but can’t think of one at the moment. I will keep this in mind for the next set of revisions.
  5. Details on the liturgy section – This article is a shortened version of my Masters Thesis where I include details like the ones you mention but I cut them here since it didn’t seem necessary. I tried implementing details back into this section, but it became too long for my preference.
  6. 365ff – The word ethnomusicologist is repeated as somewhat of a humor element. I understand your critique here in terms of detail and want more time to think about it for a future revision where I have more time to work on the piece.
  7. 415-428 – I inserted a new section into the history section (278ff) that I think fulfills this suggestion. I also adjusted the footnote – thanks for this suggestion it was super helpful!
  8. Conclusion – From that experience, I’m actually not sure if the music after the video provided space for emotional healing, but I think that there is potential for that to happen.

Reviewer 3 Report

Comments and Suggestions for Authors

This is well-written, compelling, and appropriately grounded in both theory and ethnographic detail. I like the way that this article is framed by the scene from BEEF, and your passages on generational trauma and healing are particularly strong.

A couple of notes:

At 299 please give us a footnote listing the other nine models

Section 4 could be enhanced by a couple of photographs of representative congregations at worship – for example, a typical Korean protestant meeting, the Heritage Mass (or other equivalent meeting) where more effusive gestures are used, and perhaps a photo of Glory church where the different styles are on display.

I would take the caveat you express in footnote 11 and move it into the main text, and perhaps deal with it right away, Given the very visible cultural footprint of Pentecostal and Charismatic Christian groups, praise dancing, and the popularity of Hillsong and similar styles of worship where performers and congregants are very effusive, many readers might immediately think of those more active styles of Christian performance and have the image of a stadium of worshippers with their arms raised in their mind while reading through the following passages where you discuss the more restrained style of mainstream Protestant worship. I would acknowledge right away in your main text that the exceptions exist, but then also clarify that these styles of charismatic Christian worship are the outliers in comparison to the more restrained style. It’s worth acknowledging that Hillsong and some similar movements have gained a great deal of their momentum in recent years, and the 1.5/2nd generation Asian Americans you’re describing as being part of the “Silent Exodus” had more of their formative experiences during a time in American history when A) the more restrained style of worship was even more dominant, and B) the racial politics of the day reinforced the need to appear proper and pious and restrained.  

Comments on the Quality of English Language

By and large, the writing on this article is very good. 

In the paragraph from 55-62 there are some passages that seem just slightly colloquial--I have these suggestions: 

56 “One person is deeply moved…”

59 “…for the Steven Yeuns that are de-churched, grew up in the Korean church [insert comma], left for several years, and are coming back, or in some similar situation.”

62 “How do we maintain intellectual credibility, but at the same time…”

At 165-177 I would put the anecdote that happened on that particular day in past tense. The use of present and past tense in the following paragraphs is good, though, as it becomes clear to the reader when we are talking about the events of a specific day and when we are using those events to generalize about continued patterns of behavior.

At 306, indent the block quote

Author Response

Thank you very much for taking the time to review this manuscript. Please find the detailed responses below and the corresponding revisions

General - Block quotes have been edited, all minor edits were accepted and changed accordingly.

I added a few new paragraphs after reading other reviews such as detailing the music in the Room-for-Let (278-288), use of Konglish within the congregation at Glory, (175-185) and vocal timbre (506-522)

  1. Photo comment – Unfortunately, since the church in this study is anonymized that includes the inclusion of photographs. I totally agree though and will try to incorporate this into future studies.
  2. Footnote with nine models added
  3. Adding footnote 11 to main text/acknowledging Pentecostal and Charismatic Christian groups – As I wrote in the footnote, I think that bringing this note to the main text would derail the focus of the article and make the scope too large; however, I agree that the clarification needs to be made. A different reviewer suggested that I use the term WASP earlier in my study which I believe helps in clarifying what kind of Western worshiping body that Steuernagel, Im, and I are referring to. Please let me know if that edit comes across. Additionally, I inserted a new section into the history section (278ff) that I think also works towards fulfills this suggestion.
  4. Colloquial wording lines 55-62 – cannot edit since it’s a quote. I forgot to format it as one, and that has been edited.
  5. Present/Past tense lines 165-177 – Similarly, I forgot to format this as a quote, but I am quoting my field notes where I used present tense.

Reviewer 4 Report

Comments and Suggestions for Authors

This is probably not something that needs to be said in the article, but it might be observed that viewing this article in the light of the church catholic across twenty centuries suggests it provides another example of the church trying to figure out how to express the message of God's forgiveness, grace, mercy, and love for everyone--no exceptions, lived out in a particular temporal and ethnic context. 

Author Response

Thank you very much for taking the time to review this manuscript. Please find and updated version below.

General - Block quotes have been edited, all minor edits were accepted and changed accordingly. I added a few new paragraphs after reading other reviews such as detailing the music in the Room-for-Let (278-288), use of Konglish within the congregation at Glory, (175-185) and vocal timbre (506-522)
